# Pharmacist Intention to Provide Medication Therapy Management Services in Saudi Arabia: A Study Using the Theory of Planned Behaviour

**DOI:** 10.3390/ijerph19095279

**Published:** 2022-04-26

**Authors:** Ahmed M. Alshehri, Omar S. Alenazi, Salman A. Almutairi, Ali Z. Alali, Yasser S. Almogbel, Rana E. Alonazi, Hind A. Alkhelaifi, Waleed M. Alshehri, Faisal A. Alsehli

**Affiliations:** 1Clinical Pharmacy Department, College of Pharmacy, Prince Sattam Bin Abdulaziz University, Al-Kharj 16273, Saudi Arabia; 2College of Pharmacy, Prince Sattam Bin Abdulaziz University, Al-Kharj 16273, Saudi Arabia; alonaziom@gmail.com (O.S.A.); ph.salmanateq@gmail.com (S.A.A.); ali-za-2008@hotmail.com (A.Z.A.); 3Department of Pharmacy Practice, College of Pharmacy, Qassim University, Buraidah 51452, Saudi Arabia; y.almogbel@qu.edu.sa; 4Pharmacology Department, College of Pharmacy, Prince Sattam Bin Abdulaziz University, Al-Kharj 16273, Saudi Arabia; re.alonazi@psau.edu.sa; 5Contracts Management Department, National Unified Procurement Company (NUPCO), Riyadh 11323, Saudi Arabia; hakhelaifi@nupco.com; 6Clinical Pharmacy Department, King Fahad Medical City, Riyadh 12231, Saudi Arabia; walshehri@kfmc.med.sa; 7Pharmaceutical Care Services, King Abdulaziz Medical City, Ministry of National Guard Health Affairs, Riyadh 14611, Saudi Arabia; sehlief@ngha.med.sa; 8College of Pharmacy, King Saud Bin Abdulaziz University for Health Sciences, Riyadh 14611, Saudi Arabia

**Keywords:** medication therapy management, pharmaceutical care, Saudi Arabia, theory of planned behaviour, intention, attitude

## Abstract

Medication therapy management (MTM) is provided by pharmacists and other healthcare providers, improves patient health status, and increases the collaboration of MTM providers with others. However, little is known about pharmacists’ intention to provide MTM services in Saudi Arabia. This study aimed to predict the pharmacists’ willingness in this nation to commit to providing MTM services there. This study used a cross-sectional questionnaire based on the theory of planned behaviour (TPB). The survey was distributed to 149 pharmacists working in hospital and community pharmacies. It included items measuring pharmacist attitudes, intentions, subjective norms, perceived behavioural control, knowledge about the provision of MTM services, and other sociodemographic and pharmacy practice-related items. The pharmacists had a positive attitude towards MTM services (mean = 6.15 ± 1.12) and strong intention (mean = 6.09 ± 1.15), highly perceived social pressure to provide those services (mean = 5.42 ± 1.03), strongly perceived control over providing those services (mean = 4.98 ± 1.05), and had good MTM knowledge (mean = 5.03 ± 1.00). Pharmacists who completed a pharmacy residency programme and had good knowledge of MTM services and a positive attitude towards them usually strongly intended to provide MTM services. Thus, encouraging pharmacists to complete pharmacy residency programmes and educating them about the importance and provision of MTM services will enhance their motivation to provide them.

## 1. Introduction

Medication errors are a global issue. They are defined as any errors occurring anywhere in the medication use process [1]. Their incidence rates vary with the region but most commonly occur among patients who use many different medications [2]. A study from Saudi Arabia reported that one-fifth of all primary care prescriptions contained errors [3]. Another study from Sweden reported a 42% medication error rate [4]. In certain countries/regions, ~6–7% of all inpatients experienced a medication error, and >2/3 of the latter were avoidable [5,6]. Therefore, an effective mechanism is required to reduce the risk of medication errors.

To this end, one of the most effective approaches is the provision of medication therapy management (MTM) services. MTM is defined as a group of services that optimise therapeutic outcomes for individual patients [7]. The provision of MTM is not assigned to specific medical providers. However, pharmacists are often designated as healthcare professionals qualified to provide MTM services. In 2011, >96% of all Medicare Part-D MTM programmes, which is an optional United States national health insurance programme, utilised pharmacists to provide MTM services [8]. In addition, pharmacists have been recognised by other healthcare providers for their important rule in reducing medication errors, which could be even improved by providing MTM [9,10]. For example, studies showed that pharmacists’ participation in rounding teams with physicians and other healthcare providers resulted in low medication errors, compared with rounding teams without pharmacists [11,12]. In addition, studies showed that clinical pharmacists’ medication review services decreased the number of potential drug–drug interactions and inappropriate medications [13,14]. In this manner, clinical and financial outcomes were ameliorated for both patients and the healthcare system. Moreover, MTM provision was correlated with high levels of patient satisfaction.

From a clinical perspective, pharmacists have identified and solved problems associated with unnecessary medication therapy, incorrect medication selection and/or dosing, drug–drug interactions, adverse drug reactions, and low medication adherence [15,16,17,18,19,20,21,22,23]. The rates at which medication-related problems (MRPs) were solved were in the range of 45.0–69.1% according to previous studies [8,15,20,23]. One study found that 24.7% of all MRPs were solved by educating patients whilst the balance was corrected by altering patient medication plans [16]. Pharmacists were able to identify many of these MRPs. The most common were (1) the requirement for a less expensive medication (33.3–85%) [15,18,20], (2) the need for additional medications (22–39.8%) [8,22,23,24], (3) low medication dose (19.9–26.1%) [22,23], (4) improper prescriptions (24%) [17], and (5) low adherence (9.6–31.3%) [8,16,20,22,23]. In a review of MTM claims between 2000 and 2006 from multistate MTM administration services in the United States, pharmacists provided an average of 3.2 MTM interventions per patient. The most common among these interventions involved patient education regarding medications [25]. Pharmacy students provided an average of 3.7 MTM recommendations per patient whilst undertaking MTM courses [26]. Another study reported that pharmacists detected 38,631 MRPs between 1998 and 2008. Most of these involved the requirement for additional medication [23]. Even patients who received MTM services by telephone were able to resolve more MRPs than those who received no MTM at all [18].

The provision of MTM services by pharmacists has also realised cost savings for patients. The substitution of expensive medications with cheaper ones reduces costs for both the patient and the healthcare system. The provision of medication and health management services has a higher benefit-to-cost ratio than the absence of these services [27]. In one study, patients saved an average of USD 628/year on cardiovascular- and cerebrovascular-related medical health plan expenditures [24]. In another report, asthma patients, directly and indirectly, saved averages of USD 725/year and USD 1230/year, respectively [28]. In another study, the mean cost saving of 600 pharmacists’ recommendations in regard to adjusting medications dosage and discounting unneeded medications was USD 420,155 [29]. In another study Over 10 years, pharmacist provision of MTM services has saved the healthcare system USD 2,913,815, and the total net cost was USD 2,258,302 [23]. From the perspective of a third-party payer, spending USD 1 to provide MTM services saves USD 12.15 in healthcare expenditures.

Another important MTM outcome is patient satisfaction. One study measured patients’ satisfaction with MTM using fifteen items on a five-point Likert scale (1 = strongly disagree to 5 = strongly disagree). It measured their satisfaction with the overall MTM programme they received (e.g., satisfaction with the care they received from pharmacists) and specific element of it (e.g., satisfaction with the information received regarding their treatment goals). The study showed an average patient satisfaction level of 4.0 (±0.6) [30]. Two other studies provided MTM services for 2 and 10 years, respectively, and found that >95% of all patients were satisfied with the MTM services they received [23,31]. In addition, patients were satisfied with MTM services received via telephone or videoconference [21,31].

MTM services have been provided worldwide and have had numerous positive clinical and financial effects. Hence, encouraging pharmacists to provide MTM services would benefit both patients and the healthcare system. Awareness of pharmacist attitudes, perceived behaviour, knowledge of MTM, intention to provide MTM, and the factors that affect it might benefit healthcare policymakers by helping them eliminate any hindrances to the implementation of these valuable services.

The theory of planned behaviour (TPB) model was developed by Ajzan to identify the intentions of healthcare providers to furnish pharmaceutical services [32,33,34,35,36,37]. In the United States, Herbert et al. used the TPB model to predict the intentions of pharmacists to provide MTM. They found that pharmacists had a positive intention to provide MTM services, perceived the importance of providing MTM services in the provision of pharmacy, and received training to provide MTM services. However, they found that pharmacists still had perceived difficulties in providing MTM services due to the lack of time and support to provide MTM. Pharmacists with positive attitudes towards MTM, those with peers who encouraged them to provide MTM services, and those who had positive control over MTM provision were more strongly motivated to provide MTM [38]. MTM services in the United Stated are reimbursed by healthcare insurance, so pharmacists showed a good intention to provide MTM services. Studying the intention of pharmacists to provide MTM services in countries where the services are not reimpressed is needed.

In Saudi Arabia, to our knowledge, there is no study that identified factors affecting pharmacists’ intentions to provide MTM services. Implementation of the TBP model in Saudi Arabia would help policymakers identify the factors that influence the intention of pharmacists to provide MTM services. The aims of the present study were to identify pharmacist attitudes, subjective norms, perceived behavioural control, knowledge of MTM, and other factors that influence their intention to provide MTM services.

## 2. Methods

### 2.1. Study Design, Population, and Samples

The present research was a cross-sectional online study conducted between September 2020 and March 2021. The study population comprised pharmacists currently providing pharmaceutical care services and working either in hospitals or in community pharmacies. The survey was administered in the English language and distributed by two methods to a convenient sample of private and government hospitals and community pharmacies. First, a letter was sent from the IRB requesting Ministry of Health hospital pharmacy service departments to distribute the survey link to pharmacists and encourage them to complete it. Second, the present research team distributed barcoded flyers that staff pharmacists at hospitals and community pharmacies could use to electronically complete the survey. Pharmacists who were on site were asked to complete the survey and distribute the study among their colleagues.

### 2.2. Survey Development and Administration

The study questionnaire was created based on the TPB [39], and the manual developed by Francis et al. was used as a guide in building the study questionnaire [40]. It comprised three parts. The first measured pharmacist intention, attitudes, subjective norms (SN), and behavioural perceptions regarding the provision of MTM services. The second measured pharmacist knowledge of MTM. The third identified pharmacist sociodemographic and pharmacist-related factors.

The first part of the study included the 15 main factors of the TPB. First, pharmacist intention was defined as the degree of the expected likelihood of providing MTM services to patients, and it was measured with three items on a seven-point Likert scale (1 = strongly disagree to 7 = strongly agree). The average sum of scores for the three items was the magnitude of the pharmacist’s intention to provide MTM services. Next, pharmacist attitude was defined as the degree of importance placed on the provision of MTM services, and it was measured using four items on a seven-point Likert scale (1 = harmful and 7 = beneficial; 1 = bad and 7 = good; 1 = unpleasant for me and 7 = pleasant for me; 1 = worthless and 7 = useful). The average sum of scores for the four items was the magnitude of pharmacist attitude towards the provision of MTM services. Third, pharmacist subjective norms were defined as the perceived social pressure to provide MTM services, and they were measured using four items (1 = I should not and 7 = I should), and three items (1 = strongly disagree and 7 = strongly agree) on a seven-point Likert scale. The average sum of scores for the first four items was the magnitude of the perceived social pressure of pharmacists to provide MTM services. Finally, the perceived behavioural control of the pharmacist was the perceived control over the provision of MTM services. It was measured using a seven-point Likert scale and consisted of three items (1 = strongly disagree and 7 = strongly agree) and one other item (1= difficult and 7 = easy). The average sum of scores was the magnitude of the perceived control of the pharmacists over the provision of MTM services.

The second part of the study comprised six true/false questions obtained and modified from the literature [41]. They identified pharmacist knowledge regarding the provision of MTM services. Participants who selected ‘true’ received one point, and participants who selected ‘false’ received zero. Hence, the highest possible score was six points.

The last part of the study identified pharmacist sociodemographic factors, pharmacy education, professional experience, and settings. Its sections were related to demographics such as gender, marital status, date of birth, current working geographical region, total monthly income, and nationality, pharmacy-related variables such as pharmacy education, residency, specialty, years of practical pharmacy experience, country where the most recent degree was earned, job setting, and average working hours per week, variables related to MTM services previously provided, and any relations previously receiving MTM services.

The study questionnaire was distributed to a convenience sample of 10 pharmacists working in different hospitals and community pharmacies to assess the face and content validity of the survey. Study item internal consistency reliability was measured through Cronbach’s alpha.

### 2.3. Data Analysis

Statistical analyses were performed using STATA 16. Descriptive, simple, and multivariate linear regression analyses were used to assess the study objectives. Means and frequency distributions were used to describe the study variables. Simple linear regression analysis was used to determine the associations between the independent study variables (attitude, subjective norms, behavioural control, MTM knowledge, age (year of birth), gender, marital status, current working geographical region, total monthly income, nationality, pharmacy education, residency, specialty, years of practical pharmacy experience, country where the most recent degree was earned, job setting, average working hours per week, MTM services previously provided, and relations previously receiving MTM services) and the dependent study variable (intention to provide MTM services). Only variables with *p* < 0.2 were included in the multivariate linear regression analysis to identify all factors associated with the intention to provide MTM services [42]. Variables with *p* < 0.05 were considered statistically significant. The study was approved by the Institutional Review Board (IRB) of the Ministry of Health.

## 3. Results

### 3.1. Demographics and Practice-Setting Characteristics

There were 149 participants who completed the present study. Most participants were male (67.79%) with Saudi nationality (84.56%). Their average age was 30.33 (±6.91) years, 49.66% had never been married, and 44.30% were married. About half the participants were located in the centre of Saudi Arabia. The monthly income ranges reported by 30.87% and 22.82% of the participants were between 10,001 and 15,000 Saudi Riyals and between 5000 and 10,000 Saudi Riyals, respectively.

The participants had different education levels and pharmacy-related experiences and settings. There were 46.31% of participants with a pharmacy bachelor’s degree and 40.27% with a pharmacy doctorate (PharmD). Most of them (80.54%) had earned their most recent degree in Saudi Arabia. Most participants (79.87%) did not complete a pharmacy residency programme. Only 16.78% and 3.36% completed a general and a specialised pharmacy residency programme, respectively. Over half of the participants (57.05%) were employed in hospital pharmacies, whilst the rest were employed in community pharmacies. Of the participants working in the former (*n* = 85), their pharmacy roles, settings, and hospital types differed. Of the 85 participants, 58.82% were staff pharmacists, and 24.71% were clinical pharmacists. Moreover, 45.88% of them worked in outpatient pharmacies whilst 32.89% worked in inpatient pharmacies. About one-third (35.30%) worked in primary hospitals, another third (32.95%) worked in secondary hospitals, and the remainder worked in tertiary hospitals. Of the 64 participants who worked in community pharmacies, 73.44% worked as staff pharmacists, 18.75% were pharmacy managers, and 7.81% were pharmacy supervisors. Their average number of years of experience was 4.55 (±5.43), and they worked an average of 34.75 (±21.50) hours/week (Table 1).

### 3.2. Theory of Planned Behaviour (TPB) Constructs

Participant intentions, attitudes, subjective norms (SN), and behavioural perceptions were internally consistent (>0.6; Table 2) regarding MTM service provision. All previous domains showed slightly good reliability (0.59–0.85). The following sections provided a detailed explanation of each construct.

#### 3.2.1. Intention to Provide MTM Services

Participant intention to provide MTM services was high (mean 6.09 ± 1.15). Overall, 99 participants strongly agreed that they wanted to provide MTM for their patients.

#### 3.2.2. Attitude towards Provision of MTM Services

Participants’ positive attitude levels were high. Participants believed that providing MTM for patients was beneficial (mean = 6.26 ± 1.20), good (mean = 6.20 ± 1.19), pleasant (mean = 5.97 ± 1.41), and useful (mean = 6.17 ± 1.33) for patients. The overall participant attitude towards providing MTM services was very positive (overall mean = 6.15 ± 1.12).

#### 3.2.3. Subjective Norms

Participants felt that most people who are important to them believe they should provide MTM services (mean = 5.62 ± 1.48) and expected they would provide them (mean = 5.96 ± 1.31). Participants felt that most people who are important to them compel them to provide MTM services (mean = 4.51 ± 1.91) and strongly agreed that they wanted them to provide MTM services to their patients (mean = 5.57 ± 1.33). Overall, participants perceived that there was social pressure to provide MTM services (overall mean = 5.42 ± 1.03).

#### 3.2.4. Perceived Behavioural Control over Provision of MTM Services

Participants perceived that the control over the provision of MTM services in their pharmacy setting was greater than neutrality in all the foregoing items. They were confident that they could provide MTM services (mean = 5.62 ± 1.41) and that it was easy to provide them (mean = 5.00 ± 1.40). They were neutral about whether the provision of MTM services was beyond their control (mean = 4.85 ± 1.66) or whether the provision of MTM services was entirely their decision (mean = 4.41 ± 1.74). Overall, participants felt confident in their ability to provide MTM services in their pharmacy settings (overall mean = 4.98 ± 1.05).

### 3.3. MTM Knowledge

The average level of participants’ MTM knowledge was 4.30 (±1.00) (Table 3). Most participants answered all items related to the provision of MTM services whilst half the participants believed these services were not provided by any healthcare provider.

### 3.4. Predictors of Pharmacist Intention

The linear regression showed statistically significant associations between pharmacists’ intentions to provide MTM services and attitudes, subjective norms, behavioural control, MTM knowledge, age (year of birth), current working geographical region, residency, the average number of working hours per week, and whether a relation had received prior MTM (*p* < 0.1) (Table 4). When these variables were included in the multivariate linear regression, only pharmacy residency programme, attitudes, and MTM knowledge were significant predictors of pharmacist intention to provide MTM services in the future when all other variables were fixed. Completion of a pharmacy residency programme had a positive effect on pharmacist intention to provide MTM services (β = 0.546; 95% CI = 0.208–0.885; *p* = 0.002). A positive attitude towards providing MTM services had a positive effect on the intention to provide MTM services (β = 0.603; 95% CI = 0.461–0.745; *p* < 0.001). A good knowledge of MTM had a positive effect on the intention to provide MTM services (β = 1.020; 95% CI = 0.129–1.911; *p* = 0.025).

## 4. Discussion

To the best of our knowledge, this study is the first to identify the intention of pharmacists to provide MTM services in Saudi Arabia. Pharmacists completing a residency programme, their attitudes towards MTM, and their knowledge of MTM were the only predictors of pharmacists’ intentions to provide MTM services. Pharmacists who had high levels of knowledge regarding MTM service provision, those who completed a residency programme, and those who had positive attitudes towards MTM services had comparatively stronger intentions to provide them.

Overall, pharmacists showed a strong intention to provide MTM services. In the present study, >85% of all participants indicated that they did intend to provide MTM services. By contrast, a previous study reported that only ~75% of all pharmacists surveyed intended to provide MTM services [38]. In general, the present study corroborated the findings of prior research in the sense that it showed that pharmacists generally intend to provide medication-related services [43,44] and report adverse drug reactions [37].

For the most part, the pharmacists surveyed in the present study had a positive attitude towards providing MTM services. Over 83% indicated that the provision of MTM services would be beneficial or useful to their patients. However, one study found that only 54.2% of all pharmacists believed their patients would trust them in providing MTM services [38]. Another study showed that patients did not believe they required MTM services and their confidence in the pharmacists providing MTM services focused on medication dispensing [45]. Therefore, patients must be informed that pharmacists are able and willing to address their healthcare needs and provide them with healthcare services that would improve their condition.

The pharmacists surveyed in the present study believed that people who are important to them would encourage them to provide MTM services. However, the pharmacists did not sense that they were under any pressure to provide MTM services. This finding was corroborated by prior studies which reported that comparatively few patients were even aware of the availability or provision of MTM services [45]. Moreover, the patients would not be disappointed if they did not receive MTM services [38]. The provision of MTM services is not restricted to pharmacists [6]. Thus, patients may receive MTM services from other healthcare providers. Hence, pharmacists would feel less obligated to provide these services.

The pharmacists surveyed in the present study were in partial agreement about their perceived control over the provision of MTM services. Most participants were confident that they could readily and easily provide MTM services. However, they did not generally concede that providing MTM services was entirely their decision. A previous study arrived at a similar conclusion [38]. The provision of MTM services requires collaboration among physicians, nurses, and pharmacy technicians [46,47,48], the availability of a documentation system [49,50], access to patient medical records [51,52,53], support from pharmacy managers, patient acceptance [48,54], and time to devote to the provision of MTM services. Hence, even if pharmacists are able and have the confidence to provide MTM services, they would nonetheless require additional support for its implementation [55].

Overall, the survey participants had a good knowledge of MTM services. Though MTM services were formally established after 2003 in the United States, most pharmacists already tended to provide services resembling those of MTM. The present study resembled an earlier study [41] in the sense that the participants correctly answered most of the MTM knowledge items.

### 4.1. Limitations

The present study had several limitations. First, the study was cross-sectional and measured behavioural attributes at one point in time; therefore, the results cannot show a cause-and-effect relationship. Next, the survey items were formulated according to the direct TPB methodology. Thus, it might not provide meaningful information for policymakers. Second, the survey only identified the intention of pharmacists to provide MTM services. Hence, a prospective study design will be required to analyse their actual MTM service provision. Next, the survey was anonymously completed by the participants. Therefore, there was no way to validate their responses. Finally, participants were either asked to complete the study by accessing the study online link or by scanning the study barcode, so there was no way to identify the study response rate.

### 4.2. Future Study

Future studies may identify the prevalence of providing MTM services, medication therapy review, medication reconciliation, and barriers that would prevent pharmacists from providing these beneficial services in Saudi Arabia. Additionally, knowing whether nurses or other healthcare providers provide MTM services, and how their intention to provide these services is very important. In addition, knowing the clinical and financial impact of providing MTM services on patients and the healthcare system would help increase the implementation of this service across Saudi Arabia. Identifying the percentage of schools and colleges of pharmacies currently including MTM courses in their curriculum would help increase future pharmacists’ readiness to provide MTM. Lastly, since pharmacists who completed residency programmes had a positive intention to provide MTM services, further studies need to identify the residency programme curriculum and its impact on improving pharmacists’ intentions to provide MTM services.

## 5. Conclusions

According to the survey conducted in the present study, pharmacists in Saudi Arabia (1) strongly intended to provide MTM services; (2) had a positive attitude towards providing them; (3) perceived social pressure to provide them; (4) felt that they had control over providing them; and (5) had good knowledge of them. Pharmacists who completed a pharmacy residency programme had a positive attitude towards providing MTM services, had good knowledge of MTM, and strongly intended to provide MTM services. Increasing the number of pharmacy residency programmes available to pharmacists, educating pharmacists regarding the provision of MTM services, and emphasising the benefits of providing MTM services to patients and pharmaceutical careers could strengthen the motivation of pharmacists to provide MTM services and increase their engagement in them. Pharmacy colleges and professional organisations in Saudi Arabia must target the foregoing predictive factors to increase the provision of MTM services.

## Figures and Tables

**Table 1 ijerph-19-05279-t001:** Participant sociodemographic and pharmacy practice characteristics (*n* = 149).

Variables	*n*	%
Age (years) (mean ± SD)	(30.33 ± 6.91)	
Gender		
Male	101	67.79
Female	48	32.21
Marital status		
Never married	74	49.66
Married	66	44.30
Separated	4	2.68
Widowed	3	2.01
Divorced	2	1.34
Job location in Saudi region		
Riyadh	77	51.68
Makkah	17	11.41
Al-Qassim	13	8.72
Eastern	12	8.05
Asir	6	4.03
Al-Madinah	5	3.36
Tabuk	5	3.36
Jazan	5	3.36
The North Border	4	2.68
Others ^1^	4	2.68
Total monthly income		
≤5000	31	20.81
5000–10,000	34	22.82
10,001–15,000	46	30.87
15,001–20,000	24	16.11
20,001–25,000	5	3.36
25,001–30,000	7	4.70
≥30,000	2	1.34
Highest pharmacy education level ^2^		
Bachelor’s degree	69	46.31
PharmD degree	60	40.27
Master’s degree	17	11.41
Other ^3^	3	2.01
Years of pharmacy experience (mean ± SD)	(4.55 ± 5.426)	
Completed a pharmacy residency program		
No	119	79.87
Yes, general residency	25	16.78
Yes, specialised residency	5	3.36
Nationality		
Saudi Arabian	126	84.56
Egyptian	19	12.75
Other ^4^	4	2.68
Country where latest pharmacy degree or training was obtained		
Saudi Arabia	120	80.54
Egypt	17	11.41
The United States	4	2.68
The United Kingdom	3	2.01
Other ^5^	5	3.36
Pharmacy job setting		
Community	64	42.95
Hospital	85	57.05
Community pharmacy job position		
Staff pharmacist	47	73.44
Pharmacy manager	12	18.75
Pharmacy supervisor	5	7.81
Hospital pharmacy job position ^6^		
Staff pharmacist	50	58.82
Clinical pharmacist	21	24.71
Pharmacy supervisor	6	7.06
Pharmacy Manager	4	4.71
Other ^7^	2	2.35
Hospital pharmacy setting ^8^		
Outpatient pharmacy	39	45.88
Inpatient pharmacy	24	28.24
clinical pharmacy	20	23.53
Main pharmacy	1	1.18
Healthcare institution level		
Primary healthcare institution	30	35.29
Secondary healthcare institution	28	32.94
Tertiary healthcare institution	27	31.76
Average hours of work per week (mean ± SD)	(34.75 ± 21.50)	
Previously provided MTM services		
Yes	103	69.13
No	16	30.87
Number of years providing MTM services (mean ± SD)	(2.59 ± 3.94)	
Relation received or currently receiving MTM services		
Yes	67	44.97
No	82	55.03

^1^ Three participants were from Hail, Al-Baha, and Al-Jouf. ^2^ One participant did not report a valid response. ^3^ One participant had a PhD degree, one was a pharmacy student, and one was missing data. ^4^ One participant was Syrian, one was Afghani, one was Pakistani, and one was Indian. ^5^ Two participants trained in Jordan, one in Canada, one in India, and one in Syria. ^6^ Two participants did not report a valid response. ^7^ One worked in quality improvement, one was a pharmacy intern, and two were missing data. ^8^ One participant did not report a valid response.

**Table 2 ijerph-19-05279-t002:** Descriptive statics and reliability of TPB construct (n = 149).

Items	Mean (±SD)	Frequency *n* (%) ^8^						
		1	2	3	4	5	6	7
Intention								
I expect to provide MTM services for my patients ^1^	5.92 (±1.44)	3 (2.01)	3 (2.01)	4 (2.68)	12 (8.05)	25 (16.78)	27 (18.12)	75 (50.34)
I want to provide MTM services for my patients ^1^	6.30 (±1.44)	2 (1.34)	1 (0.67)	1 (0.67)	13 (8.72)	12 (8.05)	21 (14.09)	99 (66.44)
I intend to provide MTM services for my patients ^1^	6.05 (±1.26)	0 (0)	3 (2.01)	5 (3.36)	11 (7.38)	20 (13.42)	33 (22.15)	77 (51.68)
Domain Average Total	6.09 (±1.15)	Cronbach’s alpha = 0.85						
Attitude								
Providing MTM services to patients is… ^2^	6.26 (±1.20)	0 (0)	1 (0.67)	7 (4.70)	9 (6.04)	14 (9.40)	23 (15.44)	95 (63.76)
Providing MTM services to patients is… ^3^	6.20 (±1.19)	1 (0.67)	0 (0)	4 (2.68)	10 (6.71)	24 (16.11)	19 (12.75)	91 (61.07)
Providing MTM services to patients is… ^4^	5.97 (±1.41)	3 (2.01)	2 (1.34)	2 (1.34)	18 (12.08)	19 (12.75)	25 (16.78)	80 (53.69)
Providing MTM services to patients is… ^5^	6.17 (±1.33)	0 (0)	3 (2.01)	7 (4.70)	12 (8.05)	13 (8.72)	18 (12.08)	96 (64.43)
Domain Average Total	6.15 (±1.12)	Cronbach’s α = 0.85						
Subjective norms								
Most people who are important to me think that -------------- provide MTM services for my patients ^6^	5.62 (±1.48)	3 (2.01)	1 (0.67)	8 (5.37)	24 (16.11)	25 (16.78)	28 (18.79)	60 (40.27)
It is expected of me that I will provide MTM services for my patients ^1^	5.96 (±1.31)	2 (1.34)	2 (1.34)	2 (1.34)	17 (11.41)	17 (11.41)	40 (26.85)	69 (46.31)
I feel under social pressure to provide MTM services for my patients ^1^	4.51 (±1.91)	18 (12.08)	12 (8.05)	9 (6.04)	25 (16.78)	28 (18.79)	36 (24.16)	21 (14.09)
People who are important to me want me to provide MTM services for my patients ^1^	5.57 (±1.33)	1 (0.67)	4 (2.68)	2 (1.34)	25 (16.78)	34 (22.82)	36 (24.16)	47 (31.54)
Domain Average Total	5.41 (±1.03)	Cronbach’s α = 0.61						
Perceived behavioural control								
I am confident that I could provide MTM services for my patients if I want to do so ^1^	5.62 (±1.41)	2 (1.34)	3 (2.01)	6 (4.03)	19 (12.75)	33 (22.15)	32 (21.48)	54 (36.24)
It is easy for me to provide MTM services for my patients ^7^	5.0 (±1.40)	1 (0.67)	5 (3.36)	9 (6.04)	44 (29.53)	34 (22.82)	25 (16.78)	31 (20.81)
The decision to provide MTM services for my patients is beyond my control ^1^	4.85 (±1.66)	8 (5.37)	5 (3.36)	16 (10.74)	31 (20.81)	31 (20.81)	29 (19.46)	29 (19.46)
Whether I provide MTM services for my patients is entirely my decision ^1^	4.41 (±1.74)	14 (9.40)	11 (7.38)	11 (7.38)	36 (24.16)	38 (25.5)	19 (12.75)	20 (13.42)
Domain Average Total	4.98 (±1.05)	Cronbach’s α = 0.59						

SD = standard deviation; ^1^ Response scale: 1 = strongly disagree to 7 = strongly agree; ^2^ response scale: 1 = harmful to 7 = beneficial; ^3^ response scale: 1 = bad to 7 = good; ^4^ response scale: 1 = unpleasant for me to 7 = pleasant for me; ^5^ response scale: 1 = worthless to 7 = useful; ^6^ response scale: 1 = I should not to 7 = I should; ^7^ response scale: 1 = difficult to 7 = easy; ^8^ total response percentage of each item dose not equal 100 percent due to rounding.

**Table 3 ijerph-19-05279-t003:** Participant medication therapy management (MTM) knowledge (*n* = 149).

Item	True ^1^	False ^1^
MTM is defined as a service or group of services that optimise therapeutic outcomes for individual patients	146 (97.99)	3 (2.01)
Core elements of MTM service are medication therapy review (MTR), personal medication record (PMR), medication-related action plan (MAP), intervention or referral, and documentation and follow-up	136 (91.28)	13 (8.72)
The goals of MTM services are to improve understanding of medication use, medication adherence, and detection of MRPs	132 (88.59)	17 (11.41)
Any patient who uses prescription or non-prescription medications, herbal products, or other dietary supplements could potentially benefit from MTM services	134 (89.93)	15 (10.07)
The primary role of MTM service is to facilitate adherence and disease state management	126 (84.56)	23 (15.44)
MTM can be provided by any healthcare provider	75 (50.34)	74 (49.66)
Average domain total	4.30 ( ± 1.00)	

^1^ Participants who marked items as true scored 1, whilst those who marked items as false scored 0.

**Table 4 ijerph-19-05279-t004:** Multivariate linear regression analysis of factors associated with average intention to provide MTM services (*n* = 149).

Variable	β	95% CI		*p*
		Lower	Upper	
Age	−0.005	−0.035	0.025	0.748
Current working geographical region (Riyadh vs. other)	0.180	−0.095	0.455	0.197
Pharmacy residency programme (Completed vs. none)	0.546	0.208	0.885	0.002 *
Pharmacy experience (years)	0.021	−0.017	0.059	0.282
Relation received MTM	0.012	−0.276	0.299	0.937
Hours worked per week	0.001	−0.006	0.008	0.745
Previously provided MTM	−0.541	−1.475	0.393	0.254
Attitude	0.603	0.461	0.745	<0.001 *
Subjective norms	0.029	−0.127	0.185	0.715
Perceived behavioural control	0.068	−0.066	0.202	0.317
MTM knowledge	1.020	0.129	1.911	0.025 *

MTM = medication therapy management. * Statistically significant at *p* < 0.05. Adjusted R^2^ = 0.55.

## Data Availability

The data that support the findings of this study are available from the corresponding author.

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
