# Peer review of "Pharmacist Intention to Provide Medication Therapy Management Services in Saudi Arabia: A Study Using the Theory of Planned Behaviour"

_ijerph, 2022, doi:10.3390/ijerph19095279_

Round 1
Reviewer 1 Report
Then manuscript is well written and referenced. I have only two suggestions that would improve it.
- The key word list needs editing. Specifically, Pharmacist’s intention; Pharmacist’s attitude; pharmacist’s subjective norm; Pharmacist’s perceived behaviour. These are not MeSH terms, and do not add specificity.
- The limitations section should indicate that the study was cross-sectional, and measure behavioral attributes at one point in time. In addition, it is often difficult to attribute cause and effect. Repeated measures over time would give an additional measure of durability to the findings.
Author Response
Dear Reviewer
On behalf of my coauthors, I would like to thank you for the opportunity to revise and resubmit our manuscript ijerph-1649128 entitled “Pharmacist Intention to Provide Medication Therapy Management Services in Saudi Arabia: A Study Using the Theory of Planned Behaviour”
We found your comments helpful and encouraging. It allows us to clarify our study and prove its importance to science.
Please see our response attached.
Regards
Ahmed M. Alshehri, B.Pharm., M.S., Ph.D
Clinical Pharmacy Department, College of Pharmacy, Prince Sattam bin Abdulaziz University, 3987 Al Kharj, 16273-6758 Saudi Arabia
0096115886055
Ah.alshehri@psau.edu.sa

Reviewer 2 Report
Overall, my opinion is that this manuscript is very interesting and appropriate for this Journal's readers. Some data on this important topic should be better described in the introduction (I suggested many links and references). Some important limitations in methods and results should be addressed, especially sampling and statistics should be better described. However, the results are very clearly presented. Generally, the paper has medium-high issues with the standard of writing. However, in the current form shown, I suggest a major review.
The manuscript could be strengthened by attending to the following matters:
1) GENERAL COMMENTS
Positive:
- Interesting topic
- Few data
Negative:
- Important limitations (methods)
-Selection and reporting bias
Abstract
Medication Therapy Management (MTM) is provided by pharmacists = not always. In many countries, including developed countries, MTM is provided by nurses or physicians (e.g., Austria etc.). Please modify accordingly.
Please add some limitations (e.g., study design, sampling, sample size?)
INTRODUCTION
First paragraph
Please define medication error and MTM (what pharmacist works and includes).
I would suggest that the author check the impact on medication error in cases where clinical pharmacists are important members of the multidisciplinary team on the ward.
Pharmacist participation on physician rounds and adverse drug events in the intensive care unit. JAMA 1999, 282, 267-270. & Pharmacists on rounding teams reduce preventable adverse drug events in hospital general medicine units. Arch Intern Med 2003, 163, 2014-2018.
Second paragraph
Medicare Part-D MTM programmes? Please explain.
Third paragraph
Where did pharmacists provide these interventions? (reference N#19)
Clinical pharmacists' interventions are very effective also in nursing homes settings and special institutions (in terms of medication-related problems minimizing).
I suggest the authors add some references in nursing homes and special institutions (e.g., psychiatric hospitals), where clinical pharmacists' interventions reduced medication-related problems: Positive evidence for clinical pharmacist interventions during interdisciplinary rounding at a psychiatric hospital. Please provide references in Pubmed.
Fourth paragraph
There are many differences in return-on-investment. Clinical pharmacists have provided interventions (e.g., drug-drug interaction minimizing less than the proposal for a new treatment for an untreated diagnosis). I suggest that the author add a »key reference« published by Lee and coauthors. Clinical and economic outcomes of pharmacist recommendations in a Veterans Affairs medical center. Am J Health Syst Pharm. 2002 Nov 1;59(21):2070-7. doi: 10.1093/ajhp/59.21.2070. PMID: 12434719.
Clinical pharmacists' interventions in primary care settings in Slovenia led to a lower number of medications, and return on investment was very high 5:1. Evaluation of a collaborative care approach between general practitioners and clinical pharmacists in primary care community settings in elderly patients on polypharmacy in Slovenia: a cohort retrospective study reveals positive evidence for implementation.
Last paragraph
Do pharmacists provide MTM in Saudi Arabia?
METHODS
2.1. Study design, population, and samples
How did the authors select pharmacists (e.g., selection bias)?
Did the authors calculate the necessary sample to reject a null hypothesis?
2.2. Survey Development and Administration
Did the authors validate this questionnaire?
Did the authors check all items for Cronbach’s alpha?
How do the authors invite participants?
2.3. Data Analysis
Why did the authors choose p < 0.2?
Did the authors first provide linear regression with only one independent variable?
RESULTS
3.1. Demographics and practice-setting characteristics
How many participants were omitted?
Saudi Riyals = net income for 8/h/daily shift?
3.1.1. Theory of planned behaviour (TPB) constructs
good reliability (0.59)?
3.1.3. Attitude toward the provision of MTM services
OK
3.1.4. Subjective norms
OK
3.1.5. Perceived behavioural control over the provision of MTM services
Did the pharmacists in Saudi Arabia have enough data for MTM (e.g., lab tests etc....…). The topic for discussion.
3.1.6. MTM knowledge
OK
3.17. Predictors of pharmacist intention
(p < 0.1) not defined in the methods
Did the authors calculate the number of necessary participants for linear regression?
DISCUSSION
The authors should describe how MTM is provided in Saudi Arabia and reimbursed (who provides MTM; hospital or community pharmacists?).
Limitations:
Please provide more limitations, including statistics (e.g., linear regression, sample calculation, sampling).
Future study:
It would be interesting to survey physicians and policymakers. Reimbursement is an essential part of the pilot trial to national reimbursement. Check how medication review as a part of the MTM process has been successfully reimbursed in Slovenia: »Int J Clin Pharm.
The authors should also focus on other services (medication review, medication reconciliation).
CONCLUSIONS
Ok
REFERENCES
Check references because there are two similar lists.
Author Response

(The authors gave the same response as above.)

Reviewer 3 Report
the linear regression report would be strengthened by reporting the R-square value.
Given that three of the variables were significant it might be worth re-analyzing with the three variables and see what the R-square is without the insignificant variables given that R-square usually increases with the addition of any predictor variable.
Minor writing items:
Page 3 line 147 -- is 'TBP' supposed to be 'TPB'?
Author Response

(The authors gave the same response as above.)

Reviewer 4 Report
Thank you for the opportunity to review your manuscript. Well-written outline of issues arising from Opportunities and Responsibilities in Pharmaceutical Care in Saudi Arabia.
I have read through the paper with interest with some minor issues to be addressed by the authors:
In the methods section (line 128), there is no indication of the time framework. Authors describe the research as a "cross-sectional only study" without duration. Additionally, the authors continue "approved by the Institutional Review Board (IRB) of the Ministry of Health". This reviewer suggests including this statement at the end (line 198) of the 2nd section (Methods). Finally, on page 11 of 16, with the IRB Statement (from line 390 to 390), the code of approval was "20-170E": was the research study made in 2020?
This reviewer suggests looking twice at the keywords used to describe this manuscript. These 10 concepts are composed of 27 words; maybe they are too much and difficult to be retrieved in MeSH or Emtree thesaurus.
Please, check references, as they are all duplicated (one time from page 12 to 13, and another, edited with another type of letter, from the end of page 13 to 16).
Author Response

(The authors gave the same response as above.)

Reviewer 5 Report
This manuscript provides us information about Pharmacist Intention to Provide Medication Therapy Management Services in Saudi Arabia. The topic is relevant for clinical care; however, there are several concerning’s related to the presentation of the work. The results are not clearly presented. All the manuscript must be improved.
I would like to make several suggestions for revision:
Line 80-”The substitution of cheaper medications for more expensive ones lowers costs for both the patient and the health care system”- I do not agree with this information, it's not the other way around?
Methods
Line 140-“The study questionnaire was created based on the TPB [32], and Francis et al. manual for health services researchers for construction questionnaires based on the TPB was used as a guidance in building the study questionnaire [33].”- This phrase is very confuse, please revise it.
Results
I suggest alteration of the numerations of subtitles.
For example, the subtitle 3.1.1. Theory of planned behaviour (TPB) constructs is related with another topic, so I suggest an alteration to 3.2… and consequently the other subtitle related with the table 2, to 3.2.1, 3.2.2. …. The 3.1.6 because it is related to another topic it will be 3.3 and the 3.17 will be 3.4.
Line 274-“The average for participant MTM knowledge was 5.03 (± 1.00) (Table 2)” – The authors must confirm this point because I do not understand why they wrote 5.03, I think that it is a mistake, such as the number of the table. Isn't it table 3?
Line 284-“MTM (p < 0.1) (Table 3).” Isn't it table 4? I think that it is a mistake,
References
The authors have twice the references. Attention to formatting of the references.
Author Response

(The authors gave the same response as above.)

Round 2
Reviewer 2 Report
The authors accepted all of my proposed recommendations and therefore I suggest acceptance.
Reviewer 3 Report
The authors improved the content with their edits.
Reviewer 5 Report
The authors clearly improved the manuscript, therefore, in my opinion, it is now susceptible for publication
This manuscript is a resubmission of an earlier submission. The following is a list of the peer review reports and author responses from that submission.
Round 1
Reviewer 1 Report
- The authors should add more information in the Manuscript about the sampling strategy.
- The authors should provide the total number of pharmacists who were included in the sample and the response rate of the survey.
- The authors should add more information about the survey instrument. Is the Questionnaire anonymous? what was the time for completion, was it self-administered, how was it withdrawn by the researchers?
- The authors should add more information about the score to evaluate the intention of pharmacists to provide MTM services and its coding.
- The authors should add more information about the strategy used to build the final linear regression models.
- Please check the total percentages in the tables because it does not always reach 100%.
- In the statistical analysis section, the authors should specify in detail the independent variables which were chosen to evaluate the association with the outcomes of interest.
- Are there any differences between respondents and non-respondents? if possible, include the Results in the Manuscript.
- The authors should address more thoroughly in the Discussion section the limits of this study regarding the representativeness of the sample and the methods to collect data.
Author Response
Dear reviewer
On behalf of my coauthors, I would like to thank you for the opportunity to revise and resubmit our manuscript ijerph-1465225 entitled “Pharmacist Intention to Provide Medication Therapy Management Services in Saudi Arabia: A Study Using the Theory of Planned Behaviour”
We found your comments helpful and encouraging. They allow us to clarify our study and prove its importance to science.
Regards
Ahmed M. Alshehri, B.Pharm., M.S., Ph.D
Clinical Pharmacy Department, College of Pharmacy, Prince Sattam bin Abdulaziz University, Alkarj, Riyadh, Saudi Arabia
Prince Sattam Bin Abdulaziz University-College of Pharmacy
3987 Al Kharj, 16273-6758 Saudi Arabia
0096115886055
Ah.alshehri@psau.edu.sa

Reviewer 2 Report
The introduction should be revised to establish a clearer and more compelling motivation for the study. As currently prepared, it is difficult to recognize the manuscript’s primary purpose or its contribution to the extant literature. Further, when making a case for the study, the introduction must offer a more compelling rationale for the study and its importance. The paper needs to better engage with the literature.
The points listed under results and discussion need a sentence describing what they are describing. Also more elaboration and support from research will make those connections stronger.
Clarification of the methodology is needed as to how the survey was conducted.
More scientific backing will help suggestions given for future research.
Finally, what are the practical implications based on findings of this study? More specific and realistic (substantial) implications are required. It is difficult to recognize difference from the already-preceded research. In addition, even the supplementation is necessary for research limitations and future research.
Author Response

(The authors gave the same response as above.)

Round 2
Reviewer 1 Report
.
Reviewer 2 Report
Congrats!